# Recognition of the game situation in baseball

Yasuhiro Hashimoto[1,2,3]*, Hiroshi Takahashi[3], Hiromu Nagaura[4], Shinji Yoshitake[2], Hiroki Nakata[5]

1 Department of Medical Data Intelligence, Research Center for Health-Medical Data Science, Graduate School of Medicine, Hirosaki University, Hirosaki, Aomori, Japan, 2 Faculty of Sustainable System Sciences, Osaka Metropolitan University, Osaka City, Japan, 3 Department of Health, Sport and Communication, Kobe University of Future Health Sciences, Fukusaki Town, Hyogo, Japan, 4 Advanced Teacher Professional Development Program, Hokkaido University of Education, Asahikawa City, Japan, 5 Faculty of Engineering, Nara Women's University, Nara City, Japan

* yasu88_@hotmail.co.jp

**Data Availability Statement:** The minimal data set underlying the findings of this study is publicly available and can be accessed via the EMBL-EBI BioStudies database (DOI: https://doi.org/10.6019/S-BSST1392).

## Abstract

This study examines baseball players' recognition framework of out, ball, and strike counts in baseball games and clarifies the differences in psychological perspectives between batters and pitchers. The participants were 396 players (294 batters and 102 pitchers) belonging to baseball clubs at eight universities. Participants answered 288 questions for all game situations by combining out, ball, and strike counts and runner position. The advantages for batters or pitchers were evaluated using a 7-point Likert scale (from very advantageous for batters to very advantageous for pitchers). Factor analysis indicated four significant factors (36 items): "Batter's advantage count," "Pitcher's advantage count," "2 out young count," and "0 out young count." In a direct comparison of these factors between batters and pitchers, batters were more aware of their advantage over pitchers in the factors "Batter's advantage count" and "0 out young count" and disadvantage in the "Pitcher's advantage count." Significant differences in recognition of these factors were observed between batters and pitchers. Batters were more susceptible to game situations than were pitchers. Our findings suggest that baseball players recognize several types of game situations, although not an infinite number.

## Introduction

Sports dynamics are created by multiple overlapping factors such as the score, remaining time, player abilities, and tactics, leading to enjoyment. By recognizing and interpreting the game situation, players comprehend the importance of each game scene. Judging the importance of a game situation is related to individual experience and objective indicators [1] and may be divided into important and unimportant moments within a single game. In particular, important moments likely impose significant pressure and stress on players. Numerous studies have examined the performance changes induced by sports pressure [2–4]. In many sports games, the players' performance constantly changes according to game situations. Therefore, measuring and evaluating these fluctuating data during a game is difficult.

**Funding:** This study was supported by the Japan Society for the Promotion of Science, KAKENHI Grant-in-Aid for Early-Career Scientists (19K24298, 20K19500, and 24K20555) awarded to Y.H. The funders had no role in study design, data collection and analysis, decision to publish, or preparation of the manuscript.

**Competing interests:** The authors have declared that no competing interests exist.

Research on competitive sports can be broadly divided into three categories. The first category compares the performance and results before and after the game (mainly wins and losses) using physiological and psychological indicators. The game situation is often enquired immediately after the game, and the relationship between performance and victory or defeat is often examined. For example, a study investigated the relationship between Boccia athletes' game results and their pre-competition mental state in which identity, anxiety, and self-efficacy in sports and expectations for success explained 49% of the variance in results [5]. Changes in psychological state during games have attracted the interest of many researchers. For example, a study described flow experience as "carried away" [6]. Researchers also described this phenomenon as "When performing an activity that provides challenges and requisite skills, and when both challenges and skills are high and in balance, an individual is not only enjoying the moment but is also improving their capabilities with the likelihood of learning new skills and increasing self-esteem and personal complexity" [7]. Several studies have reported the flow mechanism in various situations, such as dancing in music-based exergames [8] and video games [9–11]. However, these studies considered a single variable (e.g., game results and flow scores) and did not evaluate changes during the game. Furthermore, while reviewing the details after the game, the game results may affect judgments of the importance of the game situation.

The second research design targets the game itself, where physiological indicators and results are often collected for game analysis and big-data research. Several studies have obtained data on players' heart rates during soccer [12, 13], rugby [14], motorcycle racing [15–17], and kart racing [18]. However, because heart rate is directly affected by actual exercise, distinguishing the change in heart rate between physical and psychological factors was difficult. Furthermore, only a few studies have examined individual game situations and the specific factors (situations) that affect players' emotions.

In the third category, qualitative research design involves recalling the game, performing cognitive assessments of individual game scenes, and analyzing the data. For example, a study examined the perception of flow during a game with 28 elite athletes in seven sports [19]. Another study reported self-confidence, uninterrupted focus, and concentration in athletes when they performed well [20]. Researchers have also surveyed 16 athletes in various sports using semi-structured interviews [21]. In this research method, examining players' feelings during the game and cognitively evaluating each scene was possible. Compared to the two methods mentioned above, analyzing data directly during the game was helpful. In addition, interview data were used to measure the importance of the game situation. These data included the number of utterances and people who made similar remarks. However, in this research design, the analytical data tended to be constant, making it impossible to take advantage of the nuances and deep values that are the strengths of qualitative research.

Based on this research background, the present study aimed to elucidate the recognition of continuously changing out, ball, and strike counts in baseball games. As these variables continuously change at regular intervals, examining their stepwise changes in the game situation is excellent for organizing the subjective index of recognition. In baseball games, there is a period of non-play between plays, and the plays are divided sequentially. However, in goal-type sports (e.g., soccer, basketball, and handball), the game moves continuously. The factors, including the remaining time, score difference, and number of players sent off, determine the game situation and are intricately linked to the chances of winning. For example, when the score in soccer is 1–0, the meaning of victory depends on how much time remains. In net-type games (e.g., tennis, table tennis, and badminton), the score difference and number of points left until the game ends are important.

The present study examined each count multifacetedly by comparing the pitcher's and batter's perspectives. This examination helped understand the game situation by comprehending a single situation comprehensively. In baseball, the opposing relationship between the pitcher and the batter affects winning or losing. We analyzed changes in recognition during baseball games by recalling game situations using subjective and objective factors. The difference in characteristics between our study and the third research design is the objective factor. The present study included the out, ball, and strike counts as quantitative data. Therefore, we compared the data of the pitchers and batters and generalized them using statistical methods. Additionally, questions were based on variables that objectively represent the game situation (specifically, the number of outs, balls, and strikes). In other words, this study changed the data from a conventional research design to quantitative data and provided a structured framework for the questions. We assumed that our method could objectively investigate the effect of changes in game situations on player recognition. We hypothesized that the evaluation of the game situation would differ between pitchers and batters because they are in opposing positions. This study clarified the differences in psychological perspectives between pitchers and batters, even when in the same situation.

## Methods

### Participants

This study surveyed 544 baseball players from eight university baseball clubs using an online questionnaire (Google Forms) from April to July 2021. Of these, 396 (72.79%) players provided valid responses. The average age of the players was 19.7 ± 4.7 years, and the average game history was 12.1 ± 2.3 years. Several of the surveyed universities have produced professional baseball players. This study complied with the principles of the Declaration of Helsinki for human experimentation and was approved by the University Ethics Committee (approval number: 2021001). All the participants provided informed consent to participate in the study.

### Procedure

The participants were asked about their date of birth (age), university, dominant (throwing) hand, and perspective of response (batter and pitcher viewpoint and whether batter or pitcher is advantageous in game situations). Game situations were classified into out count (three levels), ball count (four levels), strike count (three levels), and runner position (eight levels). The participants answered 296 questions by combining all the game situations. In total, 288 questions and 8 items served as the basis for the Likert scale. In extreme cases participants may have answered the questions without reading the text, therefore analyzing these data may have yielded inaccurate results. Therefore, establishing the Likert scale ensured the accuracy of the responses before conducting the analysis. Many previous questionnaires used extreme questions to create baseline questions for the Likert scale, for example, "I have never lost a game." However, by adding an extreme question here, the participants may have realized that this question was on the Likert scale. Therefore, the Likert scale in this study asked multiple questions (nine times in total) about the situation of 0 outs, 0 balls, 0 strikes, and no runners (hereafter abbreviated as Likert scale items). This study aimed to repeat the counts by changing the out, ball, and strike counts. We checked the consistency of the responses to determine whether they were valid.

To recognize the game situation, participants were instructed to answer from seven choices (1. Very advantageous for batters, 2. Quite advantageous for batters, 3. Slightly advantageous for batters, 4. Neither, 5. Slightly advantageous for pitchers, 6. Quite advantageous for pitchers, 7. The pitcher has a great advantage). They were also asked to select the item that they thought

was most applicable. Prior to the survey, the participants were informed that all game situations were in the top of the first inning, the game score was 0–0, and to judge whether the batter or pitcher was more advantageous, the batting average of 0.250 was used. The batting average was set to 0.250 because the data in Nippon Professional Baseball (NPB) was 0.250 in 2020 [22] and 0.251 in 2021 [23].

## Analysis

For the Likert scale items, the mean value and standard deviation (SD) of the responses were calculated for each participant. The exclusion criteria for the Likert scale were typically based on a specific score threshold, either above or below. However, in this study, the Likert scale score was around the midpoint of the scale (1–7 points), which is approximately 4 points. Thus, participants with an average Likert scale score of 6 points or more or 2 points or less were excluded from the analysis as extreme responses. The same question was asked repeatedly. We considered that the reliability of the data was low for participants whose responses varied significantly for the same question. Therefore, data from participants with a standard deviation (SD) of 0.5 or more were also excluded from the analysis. Additionally, if participants gave the same response to all items (e.g., answered "4" to all questions), we evaluated that they did not recognize the game situation. However, excluding such participants using the two methods described above was difficult. Therefore, data from participants whose SD for all questions, including the Likert scale, was 0.5 or less, were excluded from the analysis.

The data on all 36 count items (a combination of out, ball, and strike counts) were subjected to an analysis of variance (ANOVA) using the factor of position (pitcher vs. batter). Ceiling and floor effects were not confirmed because all counts were included in the model. Factor analysis by the maximum likelihood method was performed on 36 items that averaged the positions of runners, and the number of factors was determined. Factor analysis by the maximum likelihood method and promax rotation was performed again. In the factor analysis, items with a factor loading of .30 or less for all factors, and those with an absolute difference in the absolute value of factor less than 0.1, were judged to have multiple loads and excluded from the analysis, and the factor analysis was repeated. Subsequently, the average value of the cognitive evaluation score of the game situation was calculated for each factor. The average value between the factors was compared using ANOVA. Additionally, we performed a two-way ANOVA using count and position factors. The data on the out, ball, and strike counts were separately analyzed because interpreting these interactions was quite complex. The Bonferroni correction was used for multiple comparisons. Values were expressed as mean ± standard deviation, and the significance level was set at $p < 0.05$. SPSS Ver. 26 for Windows (IBM) was used for the statistical analysis.

## Results

The number of valid responses in this study was 396 (of 544 respondents, 72.79% responded with 294 batters and 102 pitchers were valid). The average Likert scale items for all respondents (n = 544) was 4.12 ± 1.21, and the average SD was 0.38 ± 0.51 points. The average SD for all respondents (n = 544) and all items were 1.51 ± 0.43 points.

Table 1 lists the average values of the ball and strike counts based on the out count. All items averaged 3.50 ± 1.56 points (average 3.47 ± 1.39 points for batters and 3.50 ± 1.23 points for pitchers). ANOVAs for the mean value showed no significant differences between the pitcher and batter ($F_{(1, 394)} = 0.20$, $p > 0.05$, $\eta^2 = 0.05$), while that for the SD showed a significant difference between them ($F_{(1, 394)} = 15.27$, $p < 0.01$, $\eta^2 = 0.45$).

**Table 1. Average ball and strike counts by out count.**

| | | 0 out | | | | | 1 out | | | | | 2 out | | | |
|---|---|---|---|---|---|---|---|---|---|---|---|---|---|---|---|
| | | Ball | | | | | Ball | | | | | Ball | | | |
| | | 0 | 1 | 2 | 3 | | 0 | 1 | 2 | 3 | | 0 | 1 | 2 | 3 |
| Strike | 0 | 2.44 | 2.45 | 1.94 | 1.50 | 0 | 3.18 | 2.88 | 2.24 | 1.66 | 0 | 4.03 | 3.52 | 2.63 | 1.93 |
| | 1 | 3.10 | 3.30 | 2.48 | 2.23 | 1 | 3.66 | 3.49 | 2.83 | 2.47 | 1 | 4.48 | 4.05 | 3.21 | 2.72 |
| | 2 | 4.93 | 4.71 | 4.13 | 3.34 | 2 | 5.08 | 4.82 | 4.36 | 3.51 | 2 | 5.72 | 5.34 | 4.80 | 3.80 |

ANOVAs for the mean value showed significant effects of out count ($F$ (2, 790) = 403.14, $p < 0.01$, $\eta^2 = 0.51$), ball count ($F$ (2, 790) = 339.23, $p < 0.01$, $\eta^2 = 0.46$), strike count ($F$ (2, 790) = 554.02, $p < 0.01$, $\eta^2 = 0.58$), and runner position ($F$ (2, 790) = 412.86, $p < 0.01$, $\eta^2 = 0.51$). These data indicate that the values for out and strike counts increased with increasing counts ($p < 0.01$). However, the ball count decreased with increasing counts ($p < 0.01$, all).

Owing to the factor analysis by ball, strike, and out counts, 36 items from the four factors were extracted. The decreasing eigenvalues in order were 32.10, 28.58, 8.80, and 4.43, and the four-factor structure was considered appropriate. Therefore, the factor number was fixed at four, and factor analysis was performed using the maximum likelihood promax rotation (Table 2).

The cumulative contribution rate of these four factors was 72.17%. The first factor (F1) was named the "Batter advantage count." This factor included 1 out, 3 balls, and 1 strike. The second factor (F2) consisted of 1 out, 2 balls, 2 strikes, etc., and was named "Pitcher's advantage count." The third factor (F3) was named "2 out young count." This factor included 2 outs, 0 balls, and 0 strikes. The fourth factor (F4) was named "0 out young count." This factor consisted of 0 outs, 0 balls, 0 strikes, etc. The α coefficients for each factor were F1 (α = 0.949), F2 (α = 0.939), F3 (α = 0.863), and F4 (α = 0.860).

Table 3 shows the differences in cognitive evaluation scores between pitchers and batters. The game situation recognized by the players as the batter's advantage was 0 outs, 3 balls, and 0 strikes. However, the game situations that were recognized as the most advantageous for the pitcher were 2 outs, 0 balls, and 2 strikes. Significant differences were observed for 17 of the 36 items. The ANOVAs for the mean values of the position and the cognitive evaluation scores of the four factors showed significant interaction ($F$ (3, 1182) = 14.38, $p < 0.01$, $\eta^2 = 0.035$). Post-hoc multiple comparisons demonstrated significant differences among all groups, except between pitchers F2 and F3 when examining by factor ($p < 0.01$, all). Significant differences were observed between the pitchers and batters in F1 ($p < 0.01$), F2 ($p < 0.01$), and F4 ($p < 0.01$) (Fig 1). The factors were F2 "Pitcher advantage situation," F3 "2 out young count," F4 "0 out young count," and F1 "Batter advantage count" in the order of pitcher advantage.

## Discussion

The average of all items was 3.50 ± 1.56 points, which was lower than the middle score (Neither) and indicated a value slightly closer to the batter's advantage. The reason is that each average value of the out, ball, and strike counts used the data obtained by averaging the positions of the runners (eight situations). In baseball games, the most frequently appearing situation is without a runner. Among the eight situations for the runner's position, one situation is without a runner. A runner's presence affects the defensive position of defensive players and is more advantageous to the batter. In this study, since the eight situations were averaged without weighting, the values were closer to the batter's advantage.

We used a Likert scale that repeated the same questions and confirmed their consistency. For recurring questions, we selected 0 outs, 0 balls, and 0 strikes. The average score on this question was 4.03 ± 0.55 points. This value indicates that the perception of the game situation

**Table 2. Factor pattern matrix and interfactor correlation of cognitive evaluation of game status (maximum likelihood method after promax rotation).**

| | Item | | F1 | F2 | F3 | F4 | *M* | *SD* |
|---|---|---|---|---|---|---|---|---|
| F1. Batter advantage count | | | | | | | 2.40 | 0.72 |
| | 1 Out: 3 Ball: 1 Strike | | 0.92 | 0.14 | -0.03 | -0.13 | | |
| | 1 Out: 3 Ball: 0 Strike | | 0.87 | -0.13 | -0.07 | -0.04 | | |
| | 0 Out: 3 Ball: 1 Strike | | 0.86 | 0.18 | -0.05 | 0.00 | | |
| | 1 Out: 2 Ball: 0 Strike | | 0.85 | 0.00 | 0.00 | 0.04 | | |
| | 0 Out: 3 Ball: 0 Strike | | 0.81 | -0.06 | -0.13 | 0.04 | | |
| | 1 Out: 2 Ball: 1 Strike | | 0.76 | 0.24 | 0.07 | -0.01 | | |
| | 0 Out: 2 Ball: 0 Strike | | 0.75 | -0.03 | -0.02 | 0.20 | | |
| | 2 Out: 3 Ball: 0 Strike | | 0.73 | -0.31 | 0.17 | -0.07 | | |
| | 2 Out: 3 Ball: 1 Strike | | 0.71 | -0.10 | 0.23 | -0.13 | | |
| | 0 Out: 2 Ball: 1 Strike | | 0.70 | 0.17 | 0.01 | 0.19 | | |
| | 2 Out: 2 Ball: 0 Strike | | 0.60 | -0.36 | 0.40 | -0.02 | | |
| F2. Pitcher advantage count | | | | | | | 4.48 | 0.80 |
| | 1 Out: 1 Ball: 2 Strike | | -0.01 | 0.94 | -0.04 | 0.01 | | |
| | 1 Out: 0 Ball: 2 Strike | | -0.08 | 0.90 | -0.05 | 0.01 | | |
| | 1 Out: 2 Ball: 2 Strike | | 0.11 | 0.89 | 0.00 | -0.11 | | |
| | 0 Out: 1 Ball: 2 Strike | | -0.03 | 0.86 | -0.11 | 0.11 | | |
| | 0 Out: 2 Ball: 2 Strike | | 0.08 | 0.82 | -0.04 | 0.03 | | |
| | 0 Out: 0 Ball: 2 Strike | | -0.18 | 0.72 | -0.06 | 0.27 | | |
| | 1 Out: 3 Ball: 2 Strike | | 0.53 | 0.66 | 0.02 | -0.21 | | |
| | 0 Out: 3 Ball: 2 Strike | | 0.49 | 0.65 | -0.08 | 0.00 | | |
| | 1 Out: 1 Ball: 1 Strike | | 0.48 | 0.59 | 0.08 | 0.12 | | |
| | 2 Out: 0 Ball: 2 Strike | | -0.37 | 0.57 | 0.42 | -0.09 | | |
| | 2 Out: 1 Ball: 2 Strike | | -0.32 | 0.56 | 0.44 | -0.06 | | |
| | 2 Out: 2 Ball: 2 Strike | | -0.15 | 0.53 | 0.36 | -0.13 | | |
| F3. 2 out young count | | | | | | | 4.00 | 0.73 |
| | 2 Out: 0 Ball: 0 Strike | | 0.11 | 0.01 | 0.78 | 0.06 | | |
| | 2 Out: 1 Ball: 1 Strike | | 0.11 | 0.06 | 0.76 | 0.06 | | |
| | 2 Out: 1 Ball: 0 Strike | | 0.28 | -0.25 | 0.74 | 0.06 | | |
| | 2 Out: 0 Ball: 1 Strike | | -0.11 | 0.26 | 0.66 | 0.10 | | |
| F4. 0 out young count | | | | | | | 2.75 | 0.71 |
| | 0 Out: 0 Ball: 0 Strike | | 0.08 | -0.05 | 0.10 | 0.88 | | |
| | 0 Out: 0 Ball: 1 Strike | | -0.09 | 0.21 | 0.05 | 0.79 | | |
| | 0 Out: 1 Ball: 0 Strike | | 0.34 | -0.04 | 0.10 | 0.64 | | |
| | | F1 | | -0.14 | 0.22 | 0.40 | | |
| | | F2 | | | 0.36 | 0.33 | | |
| | | F3 | | | | 0.03 | | |

is generally neutral, in which neither the pitcher's nor the batter's advantage is inclined. Additionally, the average SD was 0.38 ± 0.51 points. SD 0.51 indicates the occurrence of the floor effect. Therefore, we reliably considered that many participants stably answered "Neither" during 0 outs, 0 balls, and 0 strikes.

However, the average SD for all items was 1.51 ± 0.43 points, which was larger than the average SD of 0.38 ± 0.51 for 0 outs, 0 balls, and 0 strikes. This data indicates that a few survey participants gave the same responses to all items. Additionally, Cronbach's α coefficient during factor analysis was 0.8 or more for all four factors. Therefore, the reliability of the Likert scale used in this study was ensured.

**Table 3. Cognitive evaluation scores for pitchers and batters.**

|  | pitcher | | batter | | |
|---|---|---|---|---|---|
|  | *M* | *SD* | *M* | *SD* | *p* |
| 0 out: 0 ball: 0 strike | 2.81 | 0.79 | 2.54 | 0.75 | ** |
| 0 out: 1 ball: 0 strike | 2.91 | 0.83 | 2.53 | 0.76 | ** |
| 0 out: 2 ball: 0 strike | 2.42 | 0.78 | 2.04 | 0.89 | ** |
| 0 out: 3 ball: 0 strike | 1.87 | 0.83 | 1.62 | 0.89 | * |
| 0 out: 0 ball: 1 strike | 3.36 | 1.01 | 3.19 | 0.92 |  |
| 0 out: 1 ball: 1 strike | 3.49 | 0.87 | 3.40 | 0.83 |  |
| 0 out: 2 ball: 1 strike | 2.81 | 0.79 | 2.60 | 0.80 | * |
| 0 out: 3 ball: 1 strike | 2.53 | 0.75 | 2.42 | 0.92 |  |
| 0 out: 0 ball: 2 strike | 4.53 | 1.19 | 4.93 | 1.18 | ** |
| 0 out: 1 ball: 2 strike | 4.38 | 1.17 | 4.75 | 1.12 | ** |
| 0 out: 2 ball: 2 strike | 3.84 | 0.96 | 4.26 | 1.12 | ** |
| 0 out: 3 ball: 2 strike | 3.25 | 0.85 | 3.47 | 0.88 | * |
| 1 out: 0 ball: 0 strike | 3.39 | 0.73 | 3.29 | 0.82 |  |
| 1 out: 1 ball: 0 strike | 3.13 | 0.74 | 3.02 | 0.79 |  |
| 1 out: 2 ball: 0 strike | 2.63 | 0.76 | 2.31 | 0.90 | ** |
| 1 out: 3 ball: 0 strike | 2.05 | 0.88 | 1.78 | 0.92 | ** |
| 1 out: 0 ball: 1 strike | 3.82 | 0.88 | 3.74 | 0.90 |  |
| 1 out: 1 ball: 1 strike | 3.63 | 0.79 | 3.57 | 0.79 |  |
| 1 out: 2 ball: 1 strike | 2.94 | 0.70 | 2.89 | 0.79 |  |
| 1 out: 3 ball: 1 strike | 2.73 | 0.79 | 2.58 | 0.81 |  |
| 1 out: 0 ball: 2 strike | 4.68 | 1.27 | 5.07 | 1.12 | ** |
| 1 out: 1 ball: 2 strike | 4.44 | 1.20 | 4.82 | 1.05 | ** |
| 1 out: 2 ball: 2 strike | 3.92 | 0.91 | 4.43 | 1.04 | ** |
| 1 out: 3 ball: 2 strike | 3.45 | 0.83 | 3.64 | 0.85 |  |
| 2 out: 0 ball: 2 strike | 5.37 | 1.05 | 5.66 | 1.01 | * |
| 2 out: 1 ball: 2 strike | 5.09 | 1.17 | 5.33 | 1.02 |  |
| 2 out: 2 ball: 2 strike | 4.48 | 1.00 | 4.80 | 0.97 | ** |
| 2 out: 3 ball: 2 strike | 3.90 | 0.86 | 3.87 | 0.81 |  |
| 2 out: 0 ball: 0 strike | 4.13 | 0.77 | 3.92 | 0.86 | * |
| 2 out: 1 ball: 0 strike | 3.67 | 0.80 | 3.48 | 0.89 |  |
| 2 out: 2 ball: 0 strike | 2.98 | 0.85 | 2.68 | 1.03 | ** |
| 2 out: 3 ball: 0 strike | 2.38 | 1.04 | 2.00 | 1.04 | ** |
| 2 out: 0 ball: 1 strike | 4.44 | 0.86 | 4.40 | 0.93 |  |
| 2 out: 1 ball: 1 strike | 4.23 | 0.82 | 4.04 | 0.82 | * |
| 2 out: 2 ball: 1 strike | 3.41 | 0.77 | 3.25 | 0.91 |  |
| 2 out: 3 ball: 1 strike | 3.04 | 0.83 | 2.86 | 0.91 |  |

* $p < 0.05$

** $p < 0.01$

The cognitive scores of the game showed significant effects of out, ball, and strike counts and runner position, which means that these factors affect the judgment of the advantages/disadvantages among baseball players. The effect size was highest in the order of runner position, strike count, ball count, and out count. Of these, the third strike is a strikeout, and the fourth ball is a walk, which is closely related to the rules of baseball games. However, increasing or decreasing the out count does not directly affect the hit results. Therefore, the effect size of the out count was likely low.

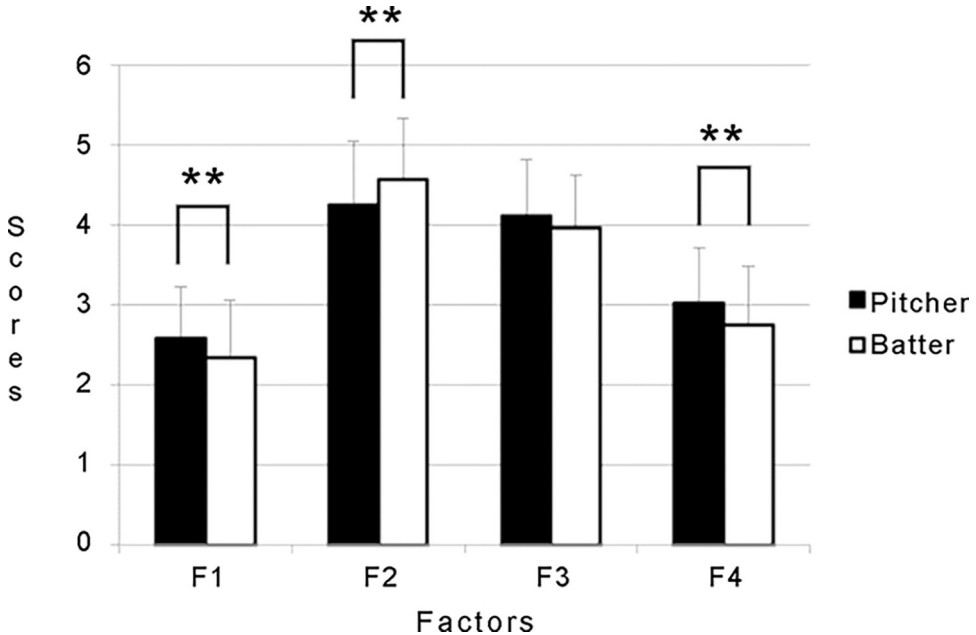

**Fig 1. Average value of cognitive evaluation by factor and viewpoint.**

A previous study used On-base plus slugging (OPS) to evaluate batting results comprehensively and indicated that the average value of all counts was 0.742 [24]. Similarly, in this study, the maximum value was 1.761 with 3 balls and 0 strikes, and the minimum value was 0.476 with 0 balls and 2 strikes. The value for 0 balls and 0 strikes was 0.934, which was 0.192 times higher than the average value. Therefore, in actual baseball games, 0 ball and 0 strike counts provide the batter with an advantage, which may differ from the player's perception. We hypothesized that baseball players misunderstand the actual advantages and disadvantages, especially for 0 balls and 0 strikes.

A study has recorded heart rates during actual baseball games and practices and investigated which factors varied among ball and strike counts and the presence or absence of runners [25]. A significant difference in heart rates was observed only for ball counts, indicating that this variable significantly affects the heart rate. Additionally, the pitcher's ball speed, rotation speed, and release point change according to the ball and strike counts [26]. Therefore, our results support previous findings and suggest that the changing environment, including ball and strike counts, is directly related to changing performance.

Owing to factor analysis, four factors (30 items) were extracted (Table 2). The characteristics of each factor are as follows: The first factor (F1) is "Batter advantage count." In this factor, 3 or 2 balls, 0 or 1 strike, and the ball counts of all items exceeded the strike counts. The average score for the first factor was 2.40 ± 0.72 points. The second factor (F2) is "Pitcher's advantage count." For this factor, all 12 factors, except 1 out, 1 ball, and 1 strike, were 2 strikes, and all items had the same number as the ball and strike counts, or the strike count was higher than the ball count. The average score of the second factor is 4.48 ± 0.80 points, while that of the entire survey is 3.50 ± 1.56 points. Based on these data, these counts are advantageous for pitchers. The third factor (F3) was "2 out young count." This factor was 2 outs for all items. The ball or strike count was 0 or 1 and does not indicate which count is more advantageous for the batter or pitcher because of the average of 4.00 ± 0.73 points. The fourth factor (F4) was "0 out young count." This factor was 0 for all items, and the ball or strike count was 0 or 1. The

fourth factor is the batter's favorable count, with an average of 2.75 ± 0.71 points. Regarding the result of the factor analysis as a whole, the cumulative contribution rate is 72.17%, and the α coefficient is 0.8 or more for all four factors. The reliability of the factor analysis in this study is high.

We excluded 2 outs, 3 balls, and 2 strikes, and 2 outs, 2 balls, and 1 strike. The common item was 2 outs, and judging the advantages or disadvantages may be difficult because they have elements of the first and second factors, which are opposite. We also excluded 1 out, 1 ball, and 0 strike; 1 out, 0 ball, and 0 strike; 0 out, 1 ball, and 1 strike; and 1 out, 0 ball, and 1 strike that have 1 out as a common item. The other "1 out young counts" were categorized as batter- or pitcher-favorable.

In this study, the three variables of out, ball, and strike counts were used as the game situation factors. The factors were divided into four categories. In this classification, the batter's advantages and disadvantages were first determined by the relationship between the ball and strike counts. The other young counts were classified using the out count. In other words, factors that could not be judged by ball and strike counts were evaluated. However, according to baseball rules, ball and strike counts strongly influence batting results, unlike the out count, which may have a limited effect on the batter's outcome. Changes between 0 and 2 outs are related to the expected score, which is the average number of points scored from 0 outs, no runners on base, 2 outs, runners on first base, etc., until the end of the inning. A study analyzed data from 373 games and showed that the expected score was lower in 2 outs than for 0 outs [27]. Thus, a game situation of 2 outs may give pitchers the psychological leeway to believe that a hit is unlikely to result in a score.

Significant differences in the perception of the game situation between pitchers and batters were observed for 17 out of the 36 items (47.22%). When this result was examined by a factor, a significant difference was found between positions (pitcher vs. batter) in the first (F1), second (F2), and fourth factors (F4) (Fig 1). The results showed that batters were more aware than pitchers that they had an advantage over pitchers in the first (F1) and fourth (F4) factors and a disadvantage over pitchers in the second factor (F2). The first (F1) and fourth (F4) factors were game situations in which the batter's advantage could be obtained, whereas the second factor (F2) was the pitcher's favorable game situation. We assume that the batter's perception is more likely affected by the game situation than the pitcher's. This notion is supported by the fact that the SD of the batter is greater than that of the pitcher. We hypothesized that this was related to the differences in perception between batters and pitchers, which were in turn based on the difference in the number of plays. Specifically, once a batter is at-bat, there is an interval of eight batters before the next at-bat. By contrast, a pitcher faces all batters continuously. The small amount of batting play in a baseball game may lead batters to be more sensitive.

We also considered that our data could not apply to and fit all cases in the game situation. For example, when the score is tied, the importance of performance increases in the ninth inning than in the first inning [1]. In other words, expectations for victory have changed, depending on inning. If the score is the same, that is. 1–0, the meaning of the situation can be very different. For instance, in the first inning, there are nine chances to turn the game around; by contrast, in the ninth inning, there is only one chance to turn the game around. It is conceivable that batters and pitchers consider the ninth inning to be more important than the first inning. For this reason, we expect that the perception of advantage or disadvantage of batters and pitchers based on outs, balls, and strike counts will fluctuate more in the ninth inning than in the first.

There were some limitations in the present study. The first was that the valid response rate was not high at 72.79%. This is due to the large number of questions in the study (296) and the repetition of similar questions. When conducting a questionnaire, it is desirable to set

questions in such a way that the valid response rate is as close to 100% as possible. However, considering the questions and content, the burden on survey participants was high, and we believe that this level of valid responses is an acceptable proportion. Second, we did not take into account some game situations and factors, such as the inning (early, middle, or late), top or bottom of the inning, number of pitches per pitcher, score, or home or away games. For example, research showed that the weighting in the probability of winning differed between the first and ninth innings [1]. Additionally, other factors such as the batter's good or bad performance may be related to the recognition of actual baseball games.

In conclusion, we repeatedly conducted cognitive evaluations of the game situation of baseball players. The baseball game situation was classified into four factors (Batter advantage count, Pitcher's advantage count, 2 out young count, and 0 out young count), and significant differences in recognizing these factors were observed between batters and pitchers. Batters were more susceptible to game situations than were pitchers. These findings indicate that baseball players differentiate game situations, not only by ball and strike counts, but also by out counts. Our findings suggest that baseball players recognize several game situations rather than an infinite number.

## Acknowledgments

We express our deep gratitude to the players, managers, and staff who participated in the survey.

## Author Contributions

**Conceptualization:** Yasuhiro Hashimoto, Hiroki Nakata.

**Data curation:** Yasuhiro Hashimoto.

**Formal analysis:** Yasuhiro Hashimoto, Hiromu Nagaura.

**Funding acquisition:** Yasuhiro Hashimoto.

**Investigation:** Yasuhiro Hashimoto, Hiroshi Takahashi.

**Methodology:** Yasuhiro Hashimoto, Hiromu Nagaura, Hiroki Nakata.

**Project administration:** Yasuhiro Hashimoto, Hiroshi Takahashi, Shinji Yoshitake.

**Supervision:** Hiroshi Takahashi, Shinji Yoshitake, Hiroki Nakata.

**Validation:** Hiromu Nagaura.

**Visualization:** Yasuhiro Hashimoto, Hiroki Nakata.

**Writing – original draft:** Yasuhiro Hashimoto, Hiroki Nakata.

**Writing – review & editing:** Yasuhiro Hashimoto, Shinji Yoshitake, Hiroki Nakata.

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
