## [Decision Letter · Decision Letter 0]

24 Jun 2024

PONE-D-24-17774Recognition of the game situation in baseballPLOS ONE

Dear Dr. Hasimoto,

Thank you for submitting your manuscript to PLOS ONE. After careful consideration, we feel that it has merit but does not fully meet PLOS ONE’s publication criteria as it currently stands. Therefore, we invite you to submit a revised version of the manuscript that addresses the points raised during the review process.

We look forward to receiving your revised manuscript.

Kind regards,

Nick Fogt

Academic Editor

PLOS ONE

 [This work was supported by JSPS KAKENHI Grant Number 20K19500.].  

4. Please expand the acronym “JSPS” (as indicated in your financial disclosure) so that it states the name of your funders in full.

Additional Editor Comments:

Two reviewers have assessed the paper. While both reviews were generally positive, there is concern over the very narrow scope of the situation assessed in the paper. Please address the potential (or lack thereof) for obtaining different results depending on the situation (eg. inning and score) in the game. Reviewer #2 also discussed the idea that uncertainty may be higher for batters than for pitchers. Please discuss this point. That is, do the authors agree with this point and how is this idea potentially reflected in the results?

Reviewers' comments:

Reviewer's Responses to Questions

**Comments to the Author**

1. Is the manuscript technically sound, and do the data support the conclusions?

Reviewer #1: Yes

Reviewer #2: Yes

2. Has the statistical analysis been performed appropriately and rigorously? 

Reviewer #1: Yes

Reviewer #2: Yes

3. Have the authors made all data underlying the findings in their manuscript fully available?

Reviewer #1: Yes

Reviewer #2: Yes

4. Is the manuscript presented in an intelligible fashion and written in standard English?

Reviewer #1: Yes

Reviewer #2: Yes

5. Review Comments to the Author

Reviewer #1: The statistical analysis in this manuscript was fascinating. I found it interesting that the author chose a hypothetical baseball game with a score of 0-0 in the 1st inning. Baseball seems to have an infinite amount of situations, but the author did a great job to break down a fraction of them and draw conclusions. Great work!

Reviewer #2: The survey is thorough, well analyzed, and has an interesting premise. However, the scope is limited in many ways. The context of a first inning, no 0-0 game score is, itself, a factor in judging pitcher or batter advantage. How different might the results be in contexts such as late innings with "your" team behind (or ahead) by one run? It is meaningful that hitters have more extreme views of pitch count advantage for both batters and pitchers. Of course, batters have much more uncertainty.

6. PLOS authors have the option to publish the peer review history of their article (what does this mean?). If published, this will include your full peer review and any attached files.

Reviewer #1: **Yes: **Jacob Terry

Reviewer #2: No

---

## [Author Response · Author response to Decision Letter 0]

11 Jul 2024

Reviewer #1: The statistical analysis in this manuscript was fascinating. I found it interesting that the author chose a hypothetical baseball game with a score of 0-0 in the 1st inning. Baseball seems to have an infinite amount of situations, but the author did a great job to break down a fraction of them and draw conclusions. Great work!

Response: Thank you very much. We are extremely grateful for your comments on our manuscript. 

Reviewer #2: The survey is thorough, well analyzed, and has an interesting premise. However, the scope is limited in many ways. The context of a first inning, no 0-0 game score is, itself, a factor in judging pitcher or batter advantage. How different might the results be in contexts such as late innings with "your" team behind (or ahead) by one run? It is meaningful that hitters have more extreme views of pitch count advantage for both batters and pitchers. Of course, batters have much more uncertainty.

Response: As a limitation of this study, we did not take into account some game situations and factors, such as the inning (early, middle, or late), top or bottom of the inning, number of pitches per pitcher, score, or home or away games. For example, a previous research showed that the weighting in the probability of winning differed between the first and ninth innings [Lindsey, 1961]. In addition, other factors such as the batter's good or bad performance may be related to the recognition of actual baseball games. These points are listed as limitations (page 18-19, lines 324-329):

“Second, we did not take into account some game situations and factors, such as the inning (early, middle, or late), top or bottom of the inning, number of pitches per pitcher, score, or home or away games. For example, research showed that the weighting in the probability of winning differed between the first and ninth innings [1]. Additionally, other factors such as the batter's good or bad performance may be related to the recognition of actual baseball games.”

Additional Editor Comments:

Two reviewers have assessed the paper. While both reviews were generally positive, there is concern over the very narrow scope of the situation assessed in the paper. Please address the potential (or lack thereof) for obtaining different results depending on the situation (eg. inning and score) in the game. Reviewer #2 also discussed the idea that uncertainty may be higher for batters than for pitchers. Please discuss this point. That is, do the authors agree with this point and how is this idea potentially reflected in

Response: We appreciate this constructive comment. We added the discussion (pages 18, lines 309-313):

“However, we also considered that our data could not apply to and fit all cases in the game situation. For example, when the score is tied, the importance of performance increases in the ninth inning than in the first inning [1]. This may lead batters to be more sensitive about their perception of the count, but pitchers may also show a similar tendency.”

In addition, we added an explanation for the reason in greater cognitive changes in batters than pitchers (pages 18, lines 313-318):

“Additionally, the difference in the number of plays might be associated with the difference in perception between batters and pitchers. That is, once a batter is in at-bat, there is an interval of eight batters before the next at-bat as batting order. In contrast, a pitcher faces all batters continuously. The small number of batting play in a baseball game may lead batters to be more sensitive.”

Additional requirements: 

Response: The address of the reference in citation No. 23 was incomplete. Therefore, we have corrected it to include the complete address (page 24, lines 413-416). 

“[23] Une N. The "most average hitter" in 2021 is... The average OPS of players who have reached the regular at-bat is .783 for Central League and .763 for Pacific League. [in Japanese; Internet]. 2021; https://news.yahoo.co.jp/byline/unenatsuki/20211209-00271866. [Accessed 2 May 2023].”

Response: All relevant data and laboratory protocols are available from the corresponding author on reasonable request (page 24, lines 411-414).

The manuscript has now been rechecked by a native English speaker.

---

## [Editor Report · Decision Letter 1]

16 Jul 2024

PONE-D-24-17774R1Recognition of the game situation in baseballPLOS ONE

Dear Dr. Hasimoto,

Thank you for submitting your manuscript to PLOS ONE. After careful consideration, we feel that it has merit but does not fully meet PLOS ONE’s publication criteria as it currently stands. Therefore, we invite you to submit a revised version of the manuscript that addresses the points raised during the review process.

We look forward to receiving your revised manuscript.

Kind regards,

Nick Fogt

Academic Editor

PLOS ONE

Journal Requirements:

Additional Editor Comments:

Thank you for your responses. Some of the wording in lines 310-318 should be modified. Please consider the following re-write:

"However, we also considered that our data may not apply to and fit all game situations. For example, when the score is tied, the importance of performance increases in the ninth inning compared to the first inning [1]. This may lead batters to be more sensitive about their perception of the count, but pitchers may also show a similar tendency. Additionally, the difference in the number of times a batter is at the plate compared to the number of batters a pitcher faces might be associated with the difference in perception between batters and pitchers. That is, once a batter finishes an at-bat, there is an interval of eight batters before the next at-bat. In contrast, a pitcher faces all batters consecutively. The small number of times a batter is at the plate in a baseball game may lead to greater sensitivity for batters than for pitchers."

Please also consider whether this is the way you wanted to state this particular thing: "For example, when the score is tied, the importance of performance increases in the ninth inning compared to the first inning [1]." That sounds as if batters care more about their performance in the ninth inning compared to the first inning. Does this actually mean that expectations change in terms of what pitches might be thrown, or does it actually mean that batters are not as "tuned in" to the count in first inning?
---

## [Author Response · Author response to Decision Letter 1]

23 Jul 2024

Response Letter

Journal Requirements:

Response: We corrected the issue where the doi codes or URL for four citations had a "." at the end, which made them unlinkable. We were unable to find the retracted paper. Of course, if there is one, I would like to correct it. Could you please let me know?

[7] Mao Y, Roberts S, Pagliaro S, Csikszentmihalyi M, Bonaiuto M. Optimal experience and optimal identity: a multinational study of the associations between flow and social identity. Front Psychol. 2016;7. doi:10.3389/fpsyg.2016.00067

[13] Panduro J, Ermidis G, Røddik L, Vigh-Larsen JF, Madsen EE, Larsen MN, et al. Physical performance and loading for six playing positions in elite female football: full-game, end-game, and peak periods. Scand J Med Sci Sports. 2022;1(Suppl. 1): 115-126. doi:10.1111/sms.13877

[22] Une N. The "Most Average Hitter" in 2020 is... In the Pacific League, the batting average, on-base percentage, and slugging percentage are one point behind the average. [in Japanese; Internet]. 2020; https://news.yahoo.co.jp/byline/unenatsuki/20201128-00209940 [Accessed 2 May 2023]. 

[23] Une N. The "most average hitter" in 2021 is... The average OPS of players who have reached the regular at-bat is .783 for Central League and .763 for Pacific League. [in Japanese; Internet]. 2021; https://news.yahoo.co.jp/byline/unenatsuki/20211209-00271866

[27] Lindsey GR. An Investigation of Strategies in Baseball. Ops Res. 1963;11: 477-501. doi:10.1287/opre.11.4.477

Additional Editor Comments:

Thank you for your responses. Some of the wording in lines 310-318 should be modified. Please consider the following re-write:

"However, we also considered that our data may not apply to and fit all game situations. For example, when the score is tied, the importance of performance increases in the ninth inning compared to the first inning [1]. This may lead batters to be more sensitive about their perception of the count, but pitchers may also show a similar tendency. Additionally, the difference in the number of times a batter is at the plate compared to the number of batters a pitcher faces might be associated with the difference in perception between batters and pitchers. That is, once a batter finishes an at-bat, there is an interval of eight batters before the next at-bat. In contrast, a pitcher faces all batters consecutively. The small number of times a batter is at the plate in a baseball game may lead to greater sensitivity for batters than for pitchers."

Please also consider whether this is the way you wanted to state this particular thing: "For example, when the score is tied, the importance of performance increases in the ninth inning compared to the first inning [1]." That sounds as if batters care more about their performance in the ninth inning compared to the first inning. Does this actually mean that expectations change in terms of what pitches might be thrown, or does it actually mean that batters are not as "tuned in" to the count in first inning?

Response: We apologize for our insufficient explanation. We consider that expectations have changed, as you pointed out. However, we think of it as an expectation for victory, not for pitch type. If the score is the same 1-0, the meaning of the situation is very different depending on the inning. For example, in the 1st inning, there are 9 chances to turn the game around; by contrast, in the 9th inning, there is only one chance to turn the game around. It is conceivable that batters and pitchers consider the 9th inning to be more important than the 1st inning. For this reason, we expect that the perception of advantage or disadvantage of batters and pitchers based on outs, balls, and strike counts will fluctuate more in the 9th inning than in the 1st inning. We added this explanation in Discussion section (page 18, lines 317-324):

“We also considered that our data could not apply to and fit all cases in the game situation. For example, when the score is tied, the importance of performance increases in the ninth inning than in the first inning [1]. In other words, expectations for victory have changed, depending on inning. If the score is the same, that is. 1–0, the meaning of the situation can be very different. For instance, in the first inning, there are nine chances to turn the game around; by contrast, in the ninth inning, there is only one chance to turn the game around. It is conceivable that batters and pitchers consider the ninth inning to be more important than the first inning. For this reason, we expect that the perception of advantage or disadvantage of batters and pitchers based on outs, balls, and strike counts will fluctuate more in the ninth inning than in the first.”

In addition, we have reconsidered the structure of the text and moved the following sentences (page 18, lines 309-314).

“We hypothesized that this was related to the differences in perception between batters and pitchers, which were in turn based on the difference in the number of plays. Specifically, once a batter is in at-bat, there is an interval of eight batters before the next at-bat as batting order. By contrast, a pitcher faces all batters continuously. The small amount of batting play in a baseball game may lead batters to be more sensitive.”

---

## [Editor Report · Decision Letter 2]

5 Aug 2024

Recognition of the game situation in baseball

PONE-D-24-17774R2

Dear Dr. Hasimoto,

We’re pleased to inform you that your manuscript has been judged scientifically suitable for publication and will be formally accepted for publication once it meets all outstanding technical requirements.

Kind regards,

Nick Fogt

Academic Editor

PLOS ONE

Additional Editor Comments (optional):

Lines 312-313: please remove the word "in" before the term "at bat", and remove the phrase "as batting order". These are not necessary in this sentence.
---

## [Editor Report · Acceptance letter]

12 Aug 2024

PONE-D-24-17774R2 

PLOS ONE

Dear Dr. Hashimoto, 

I'm pleased to inform you that your manuscript has been deemed suitable for publication in PLOS ONE. Congratulations! Your manuscript is now being handed over to our production team.

Kind regards, 

on behalf of

Dr. Nick Fogt 

Academic Editor

PLOS ONE